# Developing an enhanced 7-color multiplex IHC protocol to dissect immune infiltration in human cancers

Zhaoyu Sun[1], Richard Nyberg[1], Yaping Wu[2], Brady Bernard[1], William L. Redmond[1]*

1 Earle A. Chiles Research Institute, Providence Cancer Institute, Portland, OR, United States of America,
2 Department of Pathology, Providence Health and Services, Portland, OR, United States of America

* william.redmond@providence.org

**Data Availability Statement:** All relevant data are within the manuscript and its Supporting Information files.

## Abstract

The TSA Opal multiplex immunohistochemistry (mIHC) protocol (PerkinElmer) has been used to characterize immune infiltration in human cancers. This technique allows multiple biomarkers to be simultaneously stained in a single tissue section, which helps to elucidate the spatial relationship among individual cell types. We developed and optimized two improved mIHC protocols for a 7-color panel containing 6 biomarkers (CD3, CD8, CD163, PD-L1, FoxP3, and cytokeratin (CK)) and DAPI. The only difference between these two protocols was the staining sequence of those 6 biomarkers as the first sequence is PD-L1/CD163/CD8/CK/CD3/FoxP3/DAPI and the second sequence is FoxP3/CD163/CD8/CK/CD3/PD-L1/DAPI. By comparing PD-L1/FoxP3 staining in mIHC and singleplex PD-L1/FoxP3 staining on the adjacent slide, we demonstrated that the staining sequence does not affect the staining intensity of individual biomarkers as long as a proper antigen retrieval method was used. Our study suggests that use of an antigen retrieval buffer with higher pH value (such as Tris-EDTA pH9.0) than that of the stripping buffers (such as citrate buffer pH6.0) is helpful when using this advanced mIHC method to develop panels with multiple biomarkers. Otherwise, individual biomarkers may exhibit different intensities when the staining sequence is changed. By using this protocol, we characterized immune infiltration and PD-L1 expression in head and neck squamous cell carcinoma (HNSCC), breast cancer (BCa), and non-small cell lung cancer (NSCLC) specimens. We observed a statistically significant increase in CD3+ cell populations within the stroma of NSCLC as compared to BCa and increased PD-L1+ tumor cells in HNSCC as opposed to BCa.

## Introduction

Cancer immunotherapy, such as checkpoint blockade, represents a powerful approach for the treatment of different types of human cancer such as melanoma, non-small cell lung cancer (NSCLC), and renal cell carcinoma, among others [1–5]. In addition, the immune profile present within each patient's tumor has been used as a valuable reference for prognosis of long-term outcomes and survival across different types of cancer [6]. These traditional

**Funding:** This work was supported by a research grant from GlaxoSmithKline and the Providence Portland Medical Foundation (to WR). The funders had no role in the study design, data collection and analysis, decision to publish, or preparation of the manuscript.

**Competing interests:** WR has received research grants from Galectin Therapeutics, Bristol-Myers Squibb, Merck, GlaxoSmithKline, MiNA Therapeutics, Nektar Therapeutics, Inhibrx, Veana Therapeutics, Aeglea Biotherapeutics, Shimadzu, OncoSec, and Calibr; patents and licensing fees from Galectin Therapeutics; and has served on Scientific Advisory Boards for Nektar Therapeutics and Vesselon. ZS, RN, YW, and BB have no conflicts of interest to disclose. This does not alter our adherence to PLOS ONE policies on sharing data and materials.

(chromogenic) IHC-based approaches have provided valuable insight into the role of immune infiltration in directing subsequent clinical response to treatment. More recently, multiplexed immunohistochemistry (mIHC)-based analysis, compared with other cutting-edge technologies, has been shown to provide unique insight into the spatial relationships among cells within the complex tumor microenvironment (TME) including infiltrating immune cells, cancer cells, and stromal cells [7, 8].

Using the TSA Opal mIHC protocol [9], multiple immune biomarkers can be detected in a single tissue section through sequential staining, regardless of the species of the primary antibodies. Therefore, this protocol can overcome the hurdle of conventional mIHC, which normally uses a cocktail of primary antibodies raised in different species of animals. In this protocol, heat-treated stripping in citrate buffer pH6.0 is used to remove primary and secondary antibodies before staining the next biomarker, but not TSA-conjugated fluorescent molecules, which was demonstrated in human melanoma tissue [10]. Since antigen retrieval was found to significantly improve IHC in formalin-fixed paraffin-embedded (FFPE) tissue [11], citrate buffer pH6.0 has been widely used in heat-induced antigen retrieval (HIAR), even though Tris-HCl shows better results at higher pH [12]. The stripping ability of citrate buffer has been addressed in sequentially-stained mIHC specimens [9, 13–15]. Notably, it has caused some confusion in developing mIHC protocols using the TSA Opal kit. Indeed, few if any studies have addressed the utility of this buffer for antigen retrieval versus its use as a stripping buffer for mIHC. This is particularly critical when multiple biomarkers in a single panel require different antigen retrieval methods, such as Tris-EDTA pH9.0 versus citrate buffer pH6.0, based upon the vendor's recommendations provided with each antibody. Importantly, how to choose or balance antigen retrieval methods for an entire panel with multiple biomarkers has not been addressed.

Automated IHC has been widely used in the *in vitro* diagnostic field as it provides efficiency and consistency for IHC staining. Previous studies have shown that heating is the most important factor, in terms of antigen retrieval [16, 17]. An optimal result of HIAR is correlated with the heating temperature (T) and the heating time (t), which means the heating condition is decided by "T x t" [12, 18, 19]. To get consistent IHC results, antigen retrieval using an autostainer requires lower temperatures (<100˚C), but much longer heating times, which keeps the antigen retrieval buffer from boiling and potentially drying out the tissue section. This includes the FDA-approved Ventana PD-L1(SP263) Assay running on the VENTANA BenchMark ULTRA (https://www.accessdata.fda.gov/cdrh_docs/pdf16/p160046c.pdf) and many other clinically-approved IHC tests conducted on autostainers (https://www.atlasantibodies.com/globalassets/protocols/ihc_ventana_protocol.pdf).

Most research labs perform manual staining and HIAR using a microwave oven or pressure cooker, in which the tissue sections are immersed in a large volume of the antigen retrieval buffer, which are capable of generating higher temperatures (≥100˚C) and shorter heating times. Other considerations include the time and temperature of primary and secondary antibody incubation and signal development. For some autostainers, such as the Ventana Benchark, antibody incubation at 37˚C (rather than room temperature) is preferred. While some reports have combined automation and manual staining for mIHC, these protocols can be difficult to follow as the working conditions of the two approaches are somewhat distinct.

In the current study, we have developed two mIHC protocols with different staining sequences. In both of the protocols, Tris-EDTA pH9.0 has been used for retrieving antigens. We started from antibody validation, optimization, and staining all biomarkers sequentially to forming a multiplex panel by manual staining. These protocols maintain tissue integrity by using $H_2O_2$ to kill exogenous horse radish peroxidase (HRP) activity between two different species of primary antibodies instead of heat-treated stripping using citrate buffer pH6.0 in a

microwave oven. In addition, we validated this protocol across multiple cancer types including human head and neck squamous cell carcinoma (HNSCC), breast cancer (BCa), and non-small cell lung cancer (NSCLC) (FFPE slides) and were able to generate consistent and robust staining results for each individual biomarker across each tumor type. Together, these data provide a simple and effective method to optimize mIHC panels for assessment of immune infiltration in human cancer tissues.

## Materials and methods

### Antibody validation and optimization

Human FFPE tonsil tissue blocks were provided by Department of Pathology at Providence Portland Medical Center (Portland, Oregon). 4μm thin sections were cut on a Leica RM2235 microtome in the IHC Core Lab of the Earle A. Chiles Research Institute (Portland, Oregon). Deparaffinization of tissue sections was done through xylenes. Rehydration was done through decreasing graded alcohol. 1X Tris-EDTA (10mM Tris Base, 1mM EDTA, pH9.0) and 0.1M Sodium Citrate pH6.0 were used for retrieving antigens in a microwave oven and a hydrophobic pen was used to circle tissue sections. Endogenous peroxidase was blocked by 3% $H_2O_2$ for 15 min at room temperature (RT). Before primary antibody incubation, tissue sections were blocked with blocking/antibody diluent (ARD1001EA, PerkinElmer) for 10 min at RT. The tissue sections were incubated with anti-PD-L1 (E1L3N, Cell Signaling), anti-CD163 (MRQ-26, Roche/Ventana), anti-CD8 (SP16, Spring Biosciences), anti-cytokeratin (CK) (AE1/AE3, Dako), anti-CD3 (SP7, Genetex), and anti-FoxP3 (236A/E7, Abcam) respectively at 4˚C, overnight in a staining tray (see Table 1 for additional details). The next morning, tissue sections were washed in 1X TBST and then incubated with secondary antibody MACH2 Rb HRP-Polymer (RHRP520H, Biocare Medical) or MACH2 M HRP-Polymer (MHRP520H, Biocare Medical) in terms of the species of the primary antibody for 30min at RT. Followed by a brief wash, tissue sections were incubated with DAB (SK-4105, Vector) for about 3 min at RT. Counterstaining was done with hematoxylin (3801562, Leica) for 45 seconds followed by rinsing and bluing in flowing tap water for about 2 min. Then, tissue sections were dehydrated through increasing graded alcohol and cleared in xylenes. The slides were mounted with cytoseal 60 (8310–4, Thermo Scientific) and dried in the chemical hood.

### Opal singleplex IHC validation

Six pairs (sets) of adjacent human FFPE tonsil sections were used for comparing chromogenic and fluorescent staining in parallel. Deparaffinization of tissue sections was done through xylenes and rehydration through decreasing graded alcohol. 1X Tris-EDTA pH9.0 was used for antigen retrieval in a microwave oven. A hydrophobic pen was used to circle tissue sections and endogenous peroxidase was blocked with 3% $H_2O_2$ for 15 min at RT. Before primary antibody incubation, the tissue sections were blocked with blocking/antibody diluent (ARD1001EA) for

**Table 1. Antibodies tested for developing the mIHC protocols.**

| Antibody | Clone | Species | Titration | Fluorophore | Vendor |
|---|---|---|---|---|---|
| PD-L1 | E1L3N | Rabbit | 1:1600 | Opal690/Opal520 | Cell Signaling |
| CD163 | MRQ-26 | Mouse | 1:4 | Opal620 | Roche/Ventana |
| CD8 | SP16 | Rabbit | 1:400 | Opal650 | Spring Biosciences |
| CD3 | SP7 | Rabbit | 1:600 | Opal540 | Genetex |
| Foxp3 | 236A/E7 | Mouse | 1:400 | Opal520/Opal690 | Abcam |
| Cytokeratin | AE1/AE3 | Mouse | 1:3000 | Opal570 | Dako |

10 min at RT. Working dilutions for anti-PD-L1, anti-CD163, anti-CD8, anti-CK, anti-CD3, and anti-FoxP3 were 1:1600, 1:4, 1:400, 1:3000, 1:600, and 1:400, respectively.

For Opal fluorescent staining, tissue sections were incubated with primary antibodies at RT in a shaking stain tray. Incubation times for anti-PD-L1, anti-CD163, anti-CD8, anti-CK, anti-CD3, and anti-FoxP3 were 30, 20, 30, 20, 15, and 30 min, respectively. After a brief wash, tissue sections were incubated with MACH2 Rb HRP-Polymer (RHRP520H) or MACH2 M HRP-Polymer (MHRP520H) for 10 min at RT. A quick wash in 1X TBST was followed by incubation with Opal690 (1:200), Opal620 (1:400), Opal650 (1:200), Opal570 (1:400), Opal540 (1:200), and Opal520 (1:400), respectively, for 10 min at RT. Counter stain was done with DAPI (1 drop of DAPI solution into 0.5ml of TBST, PerkinElmer) for 5 min at RT. After a quick wash, the slides were mounted with Prolong Diamond Antifade Mountant (p36970, Thermofisher).

For chromogenic staining, tissue sections were incubated with primary antibodies at 4˚C overnight in a staining tray. Next, tissue sections were washed in 1X TBST before they were incubated with MACH2 Rb HRP-Polymer (RHRP520H) or MACH2 M HRP-Polymer (MHRP520H) for 30 min at RT. Followed by brief wash, tissue sections were incubated with DAB (SK-4105) for about 3 min at RT. Counter staining was done in hematoxylin for 45 sec followed by rinsing and bluing in flowing tap water for 2 min. Then the tissue sections were dehydrated through increasing graded alcohol and cleared in xylenes. The slides were mounted with cytoseal 60 (8310–4) and dried in the chemical hood.

For generation of the spectral library, single biomarker staining only (without DAPI) was performed on serial FFPE tonsil sections. DAPI stained only slides (tonsil) were prepared at the same time. All scale bars represent 100 μm.

## Slide imaging—Vectra 3 automated quantitative pathology system

Before scanning the 7 color mIHC slides, optimal scanning protocols were created by optimizing the exposure time for each filter cube. Each mIHC slide requires one optimized scanning protocol. The filter cubes were selected based upon the fluorophores. A 10x objective lens was used for whole slide scans, while a 20x objective lens was used for the multispectral images (MSI's). To obtain images for establishing the spectral library, snapshots were taken from each single biomarker-stained slide. The snapshots for DAPI-stained and unstained slides were taken through the DAPI filter cube. 20 MSI images/ROIs were taken from each control (tonsil) slide. 10–15 MSI images/ROIs were taken from each patient (tumor) slide.

## Quantification analysis: InForm advanced image analysis software and QuPath quantitative pathology & bioimage analysis

The spectral library was generated using the snapshots of the single biomarker-stained slides. The spectra was selected based upon the fluorophores. Composite images were generated by extracting unmixed signals using the spectral library. Snapshots of unstained slides were used for removing background caused by tissue fixation. Quantification analysis was performed by following the standard procedure which includes Tissue Segmentation, Cell Segmentation, Phenotyping, Scoring, and Export. Tissue segmentation was not applied to tonsil tissue. The threshold of biomarker intensity was determind by a Board-certified Pathologist (Dr. Yaping Wu, MD). The percentage of positive cells from each MSI image/ROI was used for the analysis and 1-way ANOVA and Linear Regression were used for statistical analysis. Whole slide quantitative analysis was performed on singleplex chromogenic and fluorescent tonsil tissue sections using QuPath Quantitative Pathology & Bioimage Analysis [16, 17, 20]. Average and Standard Deviation and Linear Regression were used for statistical analysis using Microsoft Excel.

## Multiplexed immunohistochemistry

**Tissue fixation, embedding, and sectioning.**   Human cancer tissue fixation, embedding, sectioning, and H&E staining were done using standard protocols in the Histopathology Lab at Providence Portland Medical Center. Human tonsil tissue blocks were provided by the Histopathology Lab and sectioning was done in the EACRI IHC Core Lab.

**Deparaffinization, rehydration, and antigen retrieval.**   Human FFPE tissue sections were deparaffinized with xylenes, rehydrated through graded alcohols, and rinsed with diH$_2$O and 1X TBS by following standard protocols. Antigen retrieval was performed in Tris-EDTA pH9.0 in a microwave oven for 15 min. Retrieval buffer was refilled every 5 min. The tissue sections were cooled down on the bench top for 30 min.

**Blocking endogenous enzyme and background.**   After briefly washing in diH$_2$O and 1X TBS, the tissue sections were circled with a hydrophobic pen. Endogenous HRP was quenched by incubating with 3% H$_2$O$_2$ for 15 min at RT, then tissue sections were rinsed with diH$_2$O and 1X TBS and then blocked with blocking/antibody diluent (ARD1001EA) for 10 min at RT.

**Staining.**   The incubation of primary and secondary antibodies and fluorophores were done in a slide staining tray with shaking at RT.

I. Protocol 1. Staining sequence: PD-L1, CD163, CD8, CK, CD3, FoxP3, and DAPI

<u>First staining cycle</u>

*First biomarker*: *PD-L1 staining*. Tissue sections were incubated with anti-PD-L1 (1:1600, E1L3N) for 30 min. After a brief wash with 1X TBST, slides were incubated with MACH2 Rb HRP-Polymer (RHRP520H) for 10 min. After a brief wash with 1X TBST, Opal690 (1:200) was added and incubated for 10 min.

*Quenching exogenous HRP*. After the tissue sections were briefly washed in 1X TBST, exogenous HRP was quenched by incubating with 3% H$_2$O$_2$ for 20 min at RT. After the tissue sections were rinsed with diH$_2$O, they were transferred to 1X TBST.

*Second biomarker*: *CD163 staining*. The tissue sections were incubated with anti-CD163 (1:4, MRQ-26) for 20 min. A brief wash with 1X TBST was followed by incubating with MACH2 M HRP-Polymer (MHRP520H) for 10 min. After a brief wash with 1X TBST, the sections were incubated with Opal620 (1:400) for 10 min.

*Stripping*. After a quick wash was done in 1X TBST, the stripping was performed for 15 min in 0.1M sodium citrate pH6.0 in a 24-slot plastic staining dish in a microwave oven. The stripping buffer was refilled every 5 min. The section slides were cooled down on the bench top for 30 min.

<u>Second staining cycle</u>

*Third biomarker*: *CD8 staining*. The tissue sections were blocked with blocking/antibody diluent (ARD1001EA) for 10 min at RT followed by incubating with anti-CD8 (1:400, SP16) for 30 min. A brief wash with 1X TBST was followed by incubation with MACH2 Rb HRP-Polymer (RHRP520H) for 10 min. After a brief wash with 1X TBST, the tissue sections were incubated with Opal650 (1:200) for 10 min.

*Quenching exogenous HRP*. The exogenous HRP was quenched by 3% H$_2$O$_2$ for 20 min at RT after a brief wash with 1X TBST. After the tissue sections were rinsed with diH$_2$O, they were transferred to TBST.

*Fourth biomarker*: *Anti-human CK staining*. The tissue sections were incubated with anti-human CK (1:3000, AE1/AE3) for 20 min. A brief wash with 1X TBST was followed by incubation with MACH2 M HRP-Polymer (MHRP520H) for 10 min. After a brief wash with 1X TBST, the tissue sections were incubated with Opal570 (1:400) for 10 min.

*Stripping*. After washing in 1X TBST, stripping was performed for 15 min in 0.1M sodium citrate pH6.0 in a 24-slot plastic staining dish in a microwave oven. The stripping buffer was refilled every 5 min and slides were cooled down on the bench top for 30 min.

<u>Third staining cycle</u>

*Fifth biomarker*: *CD3 staining*. The tissue sections were blocked with blocking/antibody diluent (ARD1001EA) for 10 min at RT followed by incubating with anti-CD3 (1:600) for 15 min. A brief wash with 1X TBST was followed by tissue incubation with MACH2 Rb HRP-Polymer (RHRP520H) for 10 min. After a brief wash with 1X TBST, the tissue sections were incubated with Opal540 (1:200) for 10 min.

*Quenching exogenous HRP*. After a brief wash with 1X TBST, the exogenous HRP was quenched by 3% $H_2O_2$ for 30 min at RT. After the tissue sections were rinsed with $diH_2O$, they were transferred to 1X TBST.

*Sixth biomarker*: *FoxP3 staining*. The tissue sections were incubated with anti-FoxP3 (1:400, 236A/E7) for 30 min. A brief wash with 1X TBST was followed by tissue incubation with MACH2 M HRP-Polymer (MHRP520H) for 10 min. After a brief wash with 1X TBST, the tissue sections were incubated with Opal520 (1:400) for 10 min. Then a quick wash was done in 1X TBST.

*Counterstain and coverslip mounting*. After the sections were incubated with DAPI (1 drop of DAPI solution into 0.5ml of TBST) for 5 min at RT, the slides were mounted with Prolong Diamond Antifade Mountant (p36970).

II. Protocol 2. Staining sequence: FoxP3, CD163, CD8, CK, CD3, PD-L1, and DAPI

*First biomarker*: *FoxP3 staining*. The tissue sections were incubated with anti-FoxP3 (1:400, 236A/E7) for 30 min. A brief wash with 1x TBST was followed by tissue incubation with MACH2 M HRP-Polymer (MHRP520H) for 10 min. After a brief wash with 1x TBST, the tissue sections were incubated with Opal690 (1:400) for 10 min. Then quick wash was done in 1X TBST.

*Stripping*. After a quick wash was done in 1X TBST, the stripping was performed for 15 min in 0.1M sodium citrate pH6.0 in a 24-slot plastic staining dish in a microwave oven. The stripping buffer was refilled every 5 min. The section slides were cooled down on the bench top for 30 min.

*Second biomarker*: *CD163 staining*. The tissue sections were blocked with blocking/antibody diluent (ARD1001EA) for 10 min at RT followed by incubating with anti-CD163 (1:4, MRQ-26) for 20 min. A brief wash with 1X TBST was followed by incubation with MACH2 M HRP-Polymer (MHRP520H) for 10 min. After a brief wash with 1X TBST, the tissue sections were incubated with Opal620 (1:400) for 10 min.

*Quenching exogenous HRP*. After a brief wash with 1X TBST, the exogenous HRP was quenched by 3% $H_2O_2$ for 20 min at RT. After the tissue sections were rinsed with $diH_2O$, they were transferred to 1X TBST.

*Third biomarker*: *CD8 staining*. The tissue sections were incubated with anti-CD8 (1:400, SP16) for 30 min. A brief wash with 1X TBST was followed by incubating with MACH2 Rb HRP-Polymer (RHRP520H) for 10 min. After a brief wash with 1X TBST, the tissue sections were incubated with Opal650 (1:200) for 10 min.

*Stripping*. After a quick wash was done in 1X TBST, the stripping was performed for 15 min in 0.1M sodium citrate pH6.0 in a 24-slot plastic staining dish in a microwave oven. The stripping buffer was refilled every 5 min. The section slides were cooled down on the bench top for 30 min.

*Fourth biomarker*: *CK staining*. The tissue sections were blocked with blocking/antibody diluent (ARD1001EA) for 10 min at RT followed by incubating with anti-human-CK (1:3000, AE1/AE3) for 20 min. A brief wash with 1X TBST was followed by incubating with MACH2 M HRP-Polymer (MHRP520H) for 10 min. After a brief wash with 1X TBST, the tissue sections were incubated with Opal570 (1:400) for 10 min.

*Quenching exogenous HRP.* After a brief wash with 1X TBST, the exogenous HRP was quenched by 3% $H_2O_2$ for 20 min at RT. After the tissue sections were rinsed with diH$_2$O, they were transferred to 1X TBST.

*Fifth biomarker*: *CD3 staining*. The tissue sections were incubated with anti-CD3 (1:600, SP7) for 15 min. A brief wash with 1X TBST was followed by incubating with MACH2 Rb HRP-Polymer (RHRP520H) for 10 min. After a brief wash with 1X TBST, the tissue sections were incubated with Opal540 (1:200) for 10 min.

*Stripping.* After a quick wash was done in 1X TBST, the stripping was performed for 15 min in 0.1M sodium citrate pH6.0 in a 24-slot plastic staining dish in a microwave oven. The stripping buffer was refilled every 5 min. The section slides were cooled down on the bench top for 30 min.

*Sixth biomarker*: *PD-L1 staining*. The tissue sections were incubated with anti-PD-L1 (1:1600, E1L3N) for 30 min. A brief wash with 1X TBST was followed by tissue incubating with MACH2 Rb HRP-Polymer (RHRP520H) for 10 min. After a brief wash with 1X TBST, the tissue sections were incubated with Opal520 (1:200) for 10 min.

*Counterstain and coverslip mounting.* After the tissue sections were incubated with DAPI (1 drop of DAPI solution into 0.5ml of 1X TBST) for 5 min at RT. The slides were mounted with Prolong Diamond Antifade Mountant (p36970).

## Comparison of individual biomarkers in mIHC versus singleplex–stained adjacent tissue slides

Two batches of human FFPE tonsil tissue slides were used, each containing three serial tissue slides. The middle one in each batch was used for mIHC stain by following protocol 1 and protocol 2, respectively. The other two slides in each batch were used to stain singleplex PD-L1 and FoxP3, respectively. The images were acquired with a Vectra 3.0 Automated Quantitative Pathology Imaging System and quantification analysis was done using InForm Advanced Image Analysis Software (PerkinElmer) under the guidance of a board-certified pathologist.

## Human tissue collection, processing, and quantification analysis

Primary human tumor specimens from patients with head and neck squamous cell carcinoma (HNSCC; n = 10), non-small cell lung cancer (NSCLC; n = 12), or breast cancer (BCa; n = 10) were collected following standard-of-care surgical resection at the Providence Cancer Institute. Specimens were collected following informed consent and the study was approved by the Providence Health System Regional Institutional Review Board–Oregon (IRB#06–108). Tissues were fixed and embedded using standard clinical protocols at the Histopathology Lab. 4μm-thin sections were used for staining and mIHC images were acquired using a Vectra 3.0 Automated Quantitative Pathology Imaging System. Quantification analysis was done using InForm Advanced Image Analysis Software under the guidance of a board-certified pathologist. One-way ANOVA was used to compare overall differences among the groups. All statistical analyses were performed using GraphPad Prism software (GraphPad, San Diego, CA). A p-value of <0.05 was considered significant.

## Results

### Antibody validation and balancing antigen retrieval among multiple antibodies/biomarkers

HIAR and Tris-EDTA buffer pH9.0 were recommended (based upon vendor's data sheets) for antigen retrieval of CD163, CK, and PD-L1 in human FFPE tissue sections. Using chromogenic IHC, we optimized the titrations of these three antibodies by following the

manufacturer's instructions. A 1:4 dilution for anti-CD163, 1:3000 for anti-CK, and 1:1600 for anti-PD-L1 provided consistent staining in tonsil tissue sections (Fig 1A–1C). We used 1X TBST to replace primary or secondary antibodies on the adjacent tissue sections as a control. However, HIAR with citrate buffer pH6.0, rather than Tris-EDTA pH9.0, has been recommended by the vendors for antigen retrieval of CD3, CD8, and FoxP3 in human FFPE tissue. Thus, the quality control (QC) tests and optimization of those antibodies were done using citrate buffer pH6.0 for antigen retrieval and subsequent IHC staining. Therefore, we validated each antibody by following the vendor's protocol prior to optimization. We obtained consistent staining for CD3, CD8, and FoxP3 with a 1:50 working dilution (Fig 1D, 1E and 1F).

Even though citrate buffer pH6.0 has been suggested for antigen retrieval of CD3, CD8, and FoxP3, it does not preclude the use of alternative HIAR buffers for those antibodies. Since our panel was composed of 6 biomarkers, we sought to balance the different antigen retrieval methods in order to select one that would work for all markers. Temperature, pH, and the heating time have been identified as the most important factors for heat-induced antigen retrieval in human FFPE tissue [11, 12, 19, 21]. Various methods including a microwave oven, pressure cooker, water bath, or autoclave can be used for heating the tissue and can achieve

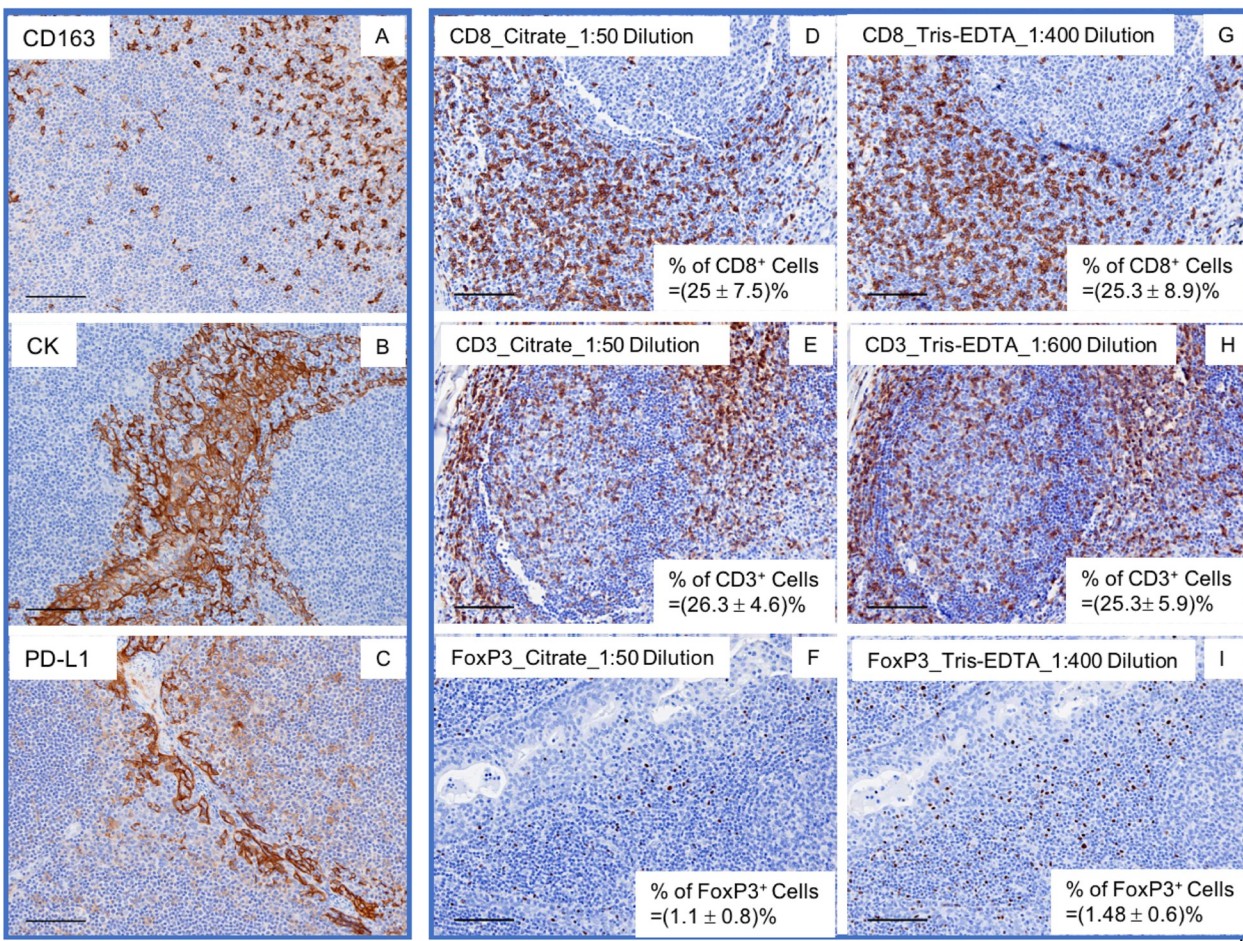

**Fig 1. Antibody validation.** Anti-CD163, CK, and PD-L1 were validated using the vendor's provided antigen retrieval method Tis-EDTA (pH 9.0) as an antigen retrieval buffer (A, B, C). Anti-CD8, CD3, and FoxP3 were validated using the vendor's suggested citrate buffer (pH 6.0) or Tris-EDTA (pH 9.0) in adjacent tissue sections (D-I). The frequency (mean+/-SD) of each biomarker was determined (D-I).

similar results on antigen retrieval by optimizing the heating time and temperature. In the current study, we used a microwave oven to heat human FFPE tissue sections.

Previous work has suggested that antigen retrieval buffers with higher pH give better results than more widely used lower pH buffers, such as citrate buffer, pH6.0 [12]. We decided to test Tris-EDTA pH9.0 on antigen retrieval of CD3, CD8, and FoxP3 epitopes. Citrate buffer pH6.0 was used for retrieving antigens in adjacent slides in parallel. We found that a 1:400 dilution works for anti-CD8 using Tris-EDTA buffer pH9.0, which was comparable with 1:50 dilution using citrate buffer pH6.0 (Fig 1D and 1G). A 1:600 dilution of anti-CD3 with Tris-EDTA buffer pH9.0 was comparable with 1:50 dilution with citrate buffer pH6.0 (Fig 1E and 1H) and a 1:400 dilution of anti-FoxP3 with Tris-EDTA buffer pH9.0 was comparable with a 1:50 dilution with citrate buffer pH6.0 (Fig 1F and 1I). To further understand these two antigen retrieval methods, we conducted quantification analysis on the entire tissue section using QuPath Quantative Pathology & Bioimage Analysis [20]. The average positive cell population and standard deviation was shown at the bottom right corner of each image (Fig 1D–1I). Together, these results suggested that Tris-EDTA buffer pH9.0 works well on retrieving those three antigens by optimizing antibody working concentrations. These data also verified that using antigen retrieval buffer with a high pH value (pH8.0–9.0) allows for robust staining with less antibody [12]. Moreover, previous work has shown that when seven different antigen retrieval solutions at different pH values ranging from 1 to 10 were compared, it was shown that the staining intensity was highest at pH8-9 among the three patterns of pH-influenced antigen retrieval staining [12]. Based on these results, Tris-EDTA buffer pH9.0 was used to retrieve all six antigens in the subsequent mIHC panels.

## Opal singleplex IHC validation

We validated and optimized the titrations of the above six antibodies using chromogenic staining in human FFPE tonsil tissue sections. Then, we tested whether the TSA Opal fluorescent stain was comparable with the corresponding chromogenic stain using 6 pairs of adjacent (consecutive) human tonsil sections to perform chromogenic and fluorescent staining in parallel. By comparing the results in adjacent/consecutive sections, we found that the singleplex Opal fluorescent stain was comparable with its chromogenic IHC stain in adjacent sections (Fig 2). To further understand these two singleplex IHC stainings, we did quantification analysis on the entire tissue sections using QuPath Quantative Pathology & Bioimage Analysis [21]. The average positive cell population and standard deviation of each biomarker was shown at the bottom right corner of each image (Fig 2). These data suggested that we could transfer the singleplex IHC to Opal mIHC panel comprised of PD-L1, CD163, CD8, CK, CD3, FoxP3, and DAPI.

## The comparison of mIHC with two different staining sequences

Since Tris-EDTA buffer pH9.0 has been tested and validated on the above-mentioned markers using the singleplex IHC stain, we used this buffer for antigen retrieval of PD-L1, CD163, CD8, CK, CD3, and FoxP3 prior to the sequential mIHC stain. This buffer was also used by others to develop a TSA-Opal mIHC panel [12, 22]. Citrate buffer pH6.0 was used to strip antibodies in developing this multiplex IHC panel with two different staining sequences on human FFPE tonsil tissue sections. To answer the question of whether the staining sequence affects the signal intensity of each biomarker in mIHC, we developed two protocols with different staining sequences: 1) PD-L1/CD163/CD8/CK/CD3/FoxP3/DAPI; and 2) FoxP3/CD163/CD8/CK/CD3/PD-L1/DAPI. The schematic diagram shows the main steps of the two protocols (Fig 3A; see Methods for detailed information). The multiplex and single-color

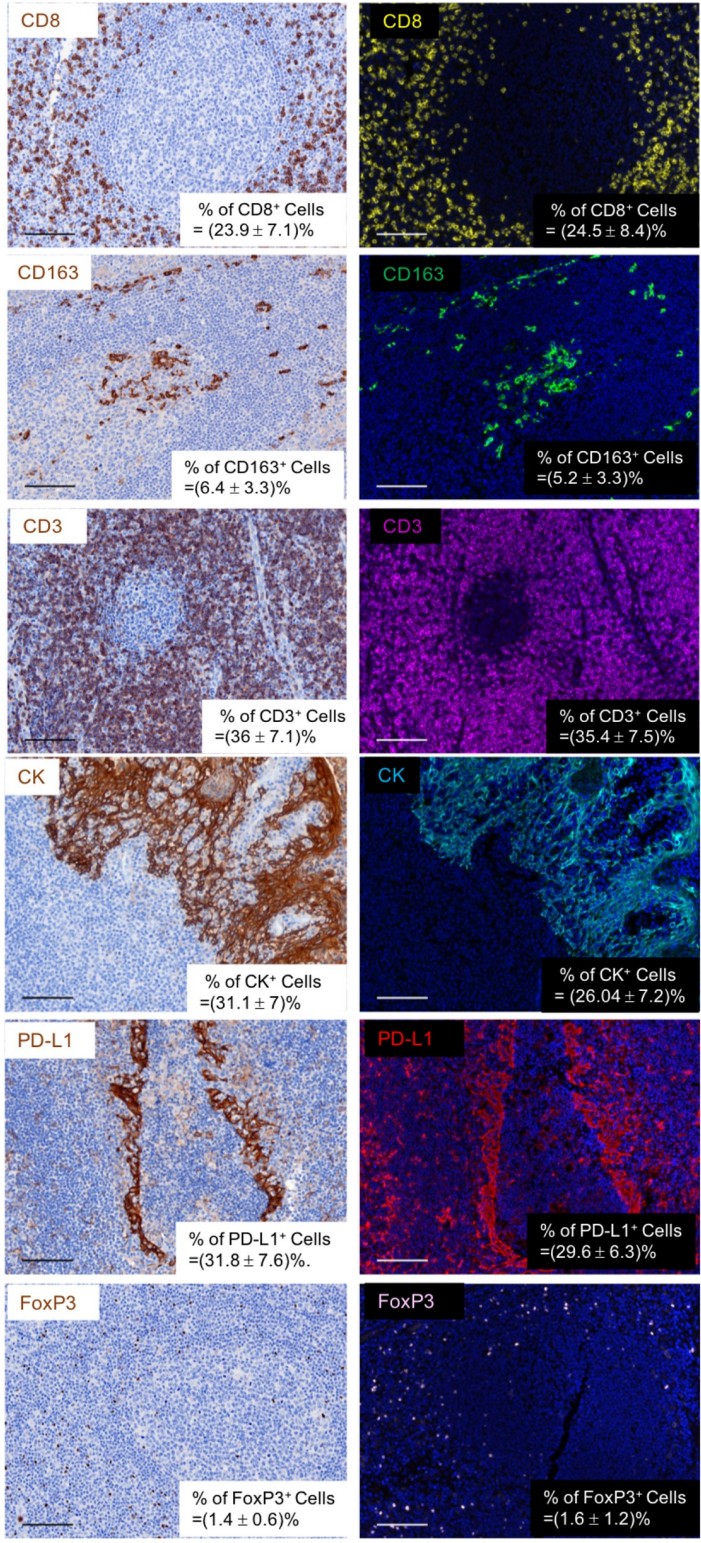

**Fig 2. Chromogenic vs. Opal: Validation of singleplex Opal fluorescent stains in human FFPE tonsil tissue specimens.** The optimized titrations of primary antibodies (CD8, CD163, CD3, FoxP3, PD-L1, and CK) which used Tris-EDTA pH9.0 to retrieve antigens in singlepex chromogenic stain were transferred to Opal fluorescent stain on the adjacent slides. The frequency (mean+/-SD) of each biomarker was determined.

images obtained from these two protocols showed specific staining with low background of each individual biomarker in human FFPE tonsil tissue sections (S1 Fig; H&E images of adjacent slides depict tissue morphology).

To dissect better the staining specificity and intensity of each individual biomarker obtained following use of the above two mIHC protocols, we analyzed PD-L1 and FoxP3 expression as these represent primarily membrane (PD-L1) and nuclear (FoxP3) staining, respectively. We found that anti-PD-L1 staining in each mIHC image (from serial sections) was comparable with its corresponding singleplex stain in the adjacent slide regardless of whether it was stained first or last in the sequential mIHC stain (Fig 3B and 3C). To further understand these two mIHC protocols, we did quantification analysis of these two biomarkers using InForm software. We chose 15 regions of interest (ROIs) in the mIHC tissue slide and 15 corresponding ROIs in adjacent singleplex IHC tissue slides following pathology review. Cell phenotyping and scoring data were used to decide PD-L1$^+$ populations due to the wide range of PD-L1 expression and the threshold of PD-L1 positivity was determined based upon pathology review. The percentage of PD-L1$^+$ cells in each corresponding ROI was used to compare between mIHC and singleplex PD-L1 stains. We found the percent of PD-L1$^+$ population obtained in 15 ROIs in mIHC slide following staining with Protocol 1 was correlated with the corresponding ROIs in singleplex PD-L1 slide. Linear Regression analysis was shown a significant correlation (Fig 4A and 4B). Similar analysis was done in another pair of mIHC by following Protocol 2 and singleplex IHC tissue slides and the percent PD-L1$^+$ population obtained with mIHC was also very similar to the corresponding ROIs in the singleplex PD-L1 slide (Fig 4C and 4D).

We conducted a similar comparison of FoxP3 expression. When FoxP3 was stained first in the mIHC panel, the percent positive cell population was correlated with the corresponding ROIs in the consecutive singleplex IHC tissue slide (Fig 4F), as we observed for PD-L1

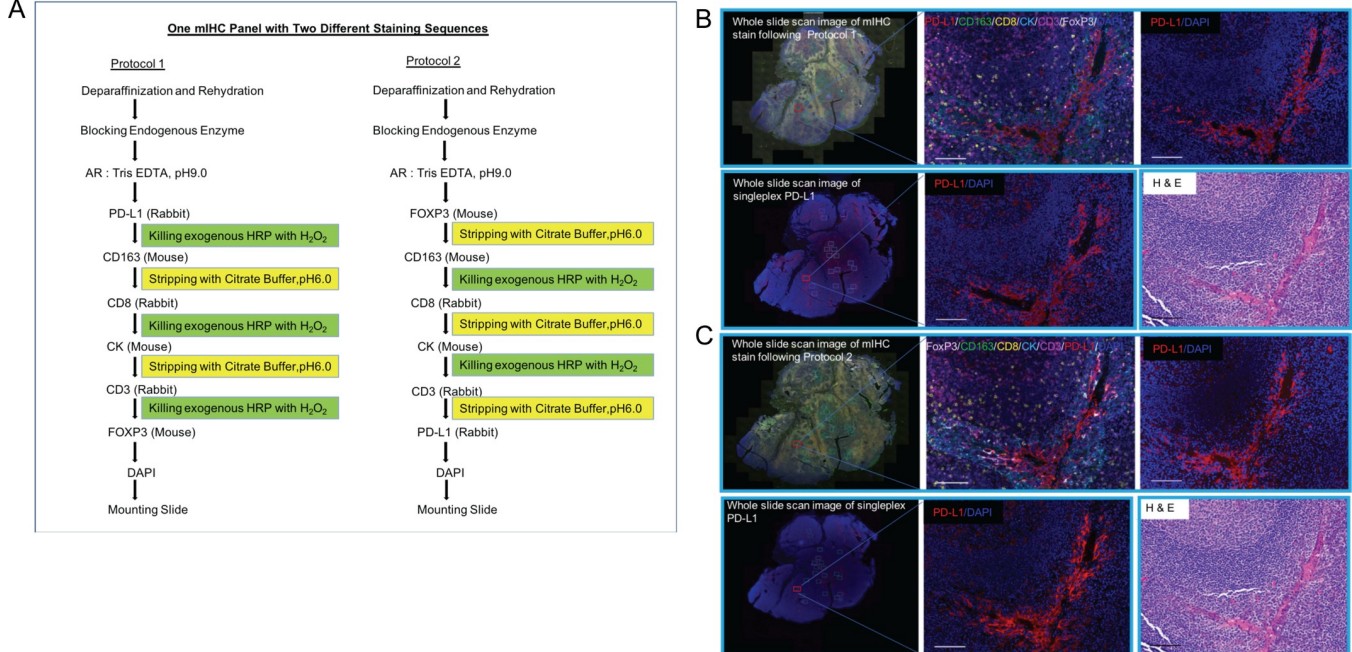

**Fig 3. Comparison of staining sequences for mIHC.** Schematic diagrams show two protocols with different staining sequences (A). Multiplex IHC and single color images were obtained using these two protocols from human tonsil (FFPE) tissue serial sections. H&E images show the morphology (B, C). Singleplex IHC in adjacent slides verified individual biomarker stains in mIHC (B, C). The mIHC stain followed Protocol 1 in (B) and Protocol 2 in (C).

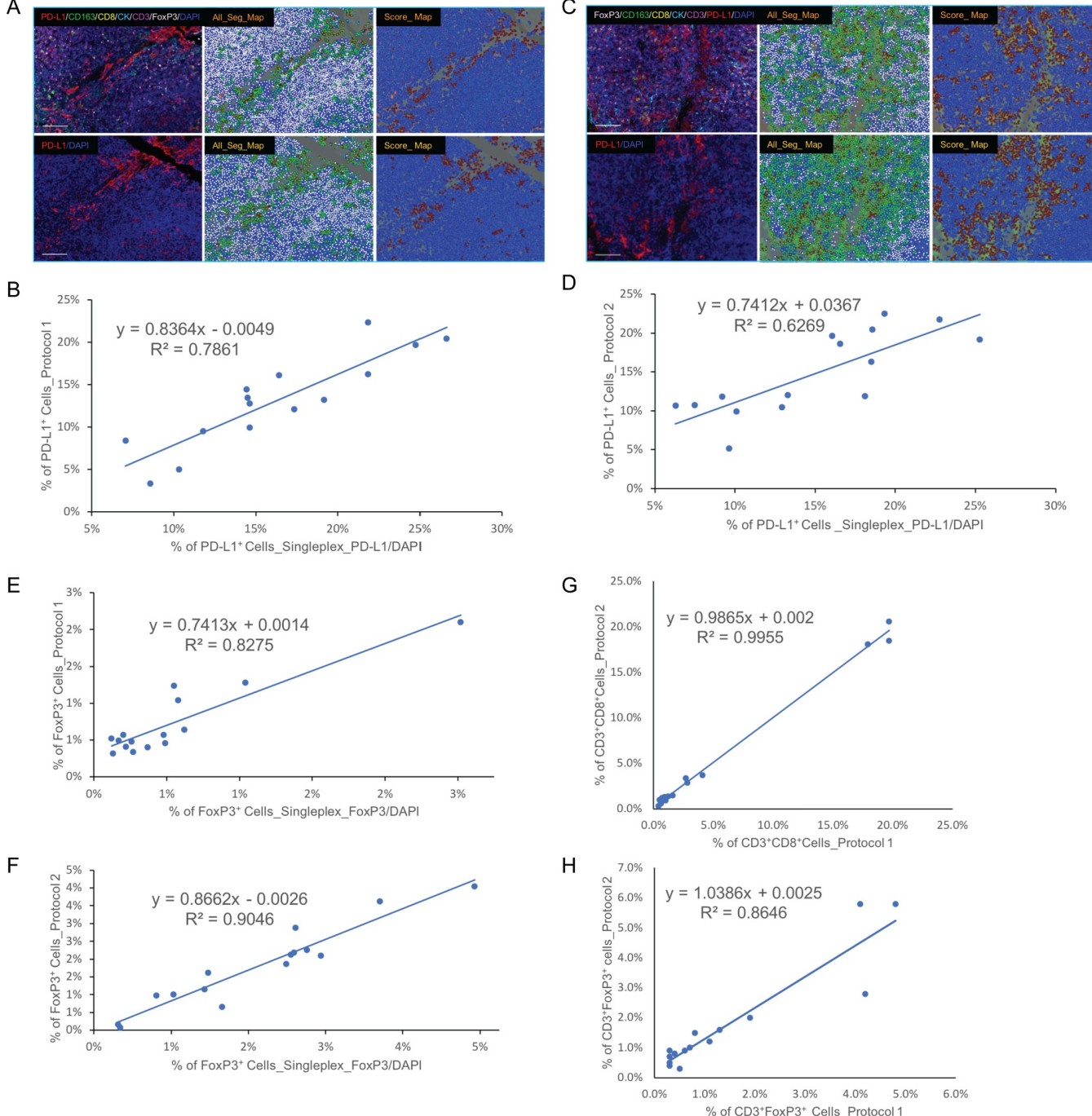

**Fig 4. Quantification analysis on PD-L1$^+$, FoxP3$^+$, CD3$^+$CD8$^+$, and CD3$^+$FoxP3$^+$ cell populations.** Quantification analysis confirmed the percent PD-L1$^+$ cell population determined by mIHC is comparable with singleplex IHC in the adjacent sections (A-D). The threshold of PD-L1$^+$ cell populations was determined by a Board-certificated Pathologist. The staining intensity of PD-L1 cells above the threshold were recognized as positive cells. The mIHC stain followed Protocol 1 in (A) and Protocol 2 in (C). Similar analysis was conducted with respect to FoxP3 expression (E, F). CD3$^+$CD8$^+$ and CD3$^+$FoxP3$^+$ cell populations stained by Protocol 1 and Protocol 2 were analyzed (G, H).

staining. When FoxP3 was stained last in the sequential mIHC, the percent positive cell population was correlated with the corresponding ROIs of the singleplex FoxP3 slide as well (Fig 4E). Together, these data suggested that the staining sequence does not affect the staining

intensity of each individual marker in this mIHC panel as long as Tris-EDTA buffer pH9.0 was suitable for antigen retrieval and citrate buffer pH6.0 as a stripping buffer. CD3$^+$CD8$^+$ and CD3$^+$FoxP3$^+$ cell populations were quantified, respectively, to determine the extent to which the staining sequence affects detection of these epitopes. Linear Regression Analysis data was shown the highly significant correlation between these two protocols (Fig 4G and 4H), demonstrating that the staining sequence does not affect detection of these populations.

## Evaluation of mIHC staining and extent of tumor infiltrating lymphocytes (TIL) in breast cancer, head and neck cancer, and non-small cell lung cancer tissue specimens

After we optimized the mIHC protocols on human FFPE tonsil tissue specimens, we tested this panel across several different cancer types. The profile of tumor infiltrating lymphocytes (TILs) and the immune checkpoint inhibitor PD-L1 was examined in head and neck squamous cell carcinoma (HNSCC), breast cancer (BCa), and non-small cell lung cancer (NSCLC) specimens obtained from primary surgical resections. We stained NSCLC (n = 15), HNSCC (n = 10), and BCa (n = 12) specimens with Protocol 1 (see representative mIHC images and H&E images of adjacent slides in Fig 5A and Table 2). Quantification analysis was done on ROIs (n = 10/slide) and different cell populations including CD3$^+$, CD8$^+$, CD163$^+$, FoxP3$^+$, PD-L1$^+$, and CK$^+$ were analyzed in stromal (CK$^-$) and tumor (CK$^+$) areas, respectively. The median from all ROIs per patient was used for comparison cross tumor types. This analysis revealed a statistically significant increase in the CD3$^+$ population within the stromal area of NSCLC as compared to BCa (Fig 5B), which reflects the low level of immune infiltration typically observed in ER$^+$, PR$^+$, and/or HER2$^+$ breast cancer [23–25]. No significant differences were observed with respect to CD8$^+$ T cells, CD163$^+$ macrophages, or FoxP3$^+$ regulatory T cell infiltration across the tumor types (Fig 5B). Further analysis revealed significantly greater PD-L1 expression within CK$^+$ tumor cells, but not stroma, in HNSCC tissue as compared to BCa (Fig 5C). Taken together, these data demonstrate the utility of this panel for investigation of immune infiltration of several different primary human tumors.

## Discussion

Conventional mIHC involves incubation of antibody cocktails on a single tissue slide, which requires those antibodies to be raised in different species of animals. Due to this hurdle, it has been very difficult to get robust staining of multiple biomarkers within a single tissue section using traditional methods. The Opal mIHC method [9] uses tyramide, which covalently binds to the same protein of the antigen or the nearby proteins, to amplify the signal. The primary and secondary antibodies can then be stripped away by heating the slides in citrate buffer pH6.0 in a microwave oven. This makes similar species of antibodies amenable for sequential staining of a single tissue section.

However, to ensure proper staining for each biomarker in a multiplex IHC panel, it is crucial to choose the appropriate antigen retrieval buffer/method for the majority of or all biomarkers within the panel. This is especially important in mIHC panels where different antigen retrieval buffers/methods are recommended by the vendor for each of the antibodies. The first issue to consider is choosing the proper antigen retrieval method. The pH value is more critical than the chemical composition of the antigen retrieval buffer solution [12]. Indeed, antigen retrieval buffers with higher pH value (such as Tris-EDTA pH9.0) are more advantageous than citrate buffer pH6.0 [12]. In our study, this was verified with the staining of CD8, CD3, and FoxP3 (Fig 1G–1I).

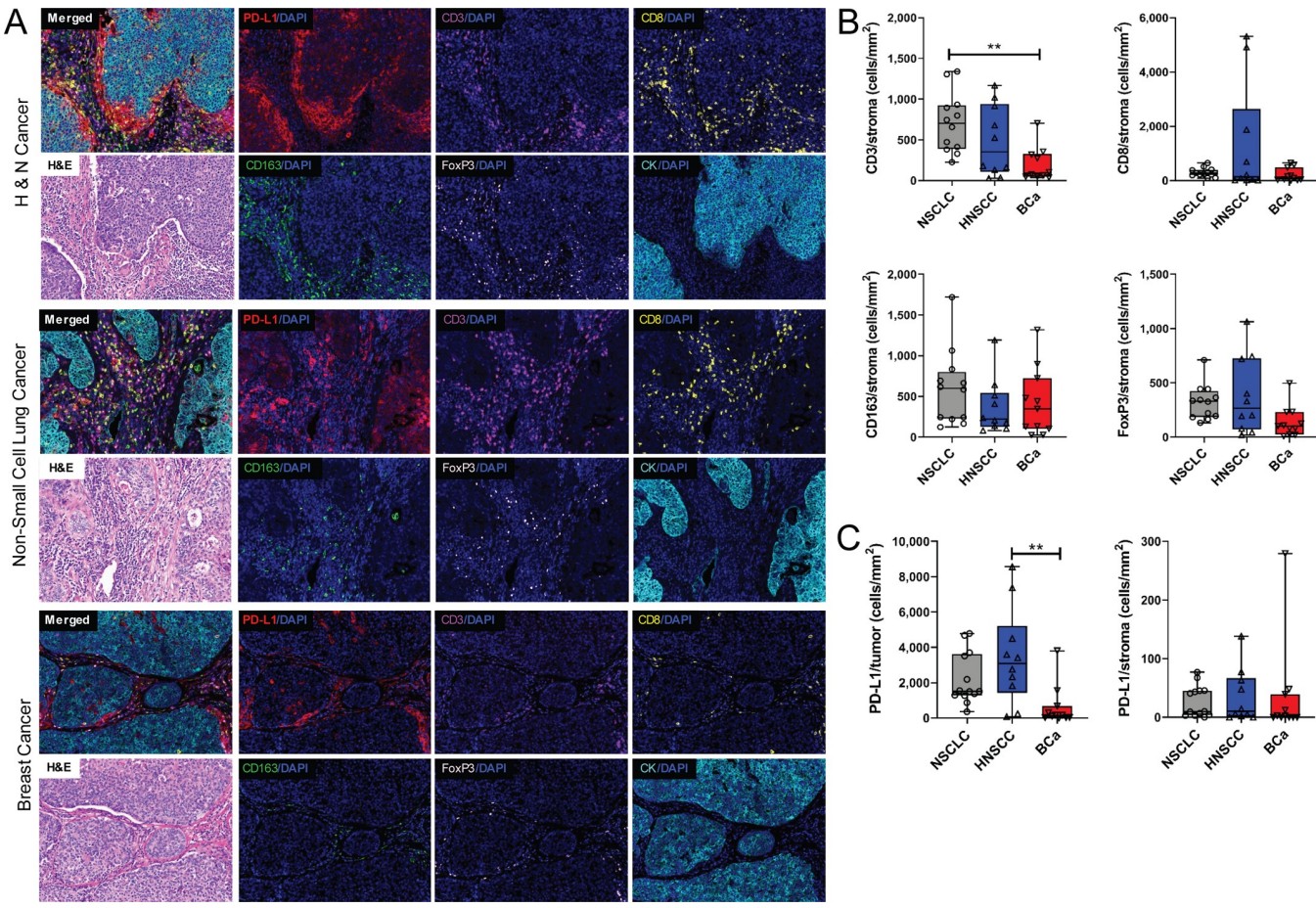

**Fig 5. mIHC staining and analysis of immune infiltration of HNSCC, NSCLC, and BCa specimens.** The protocol was validated in head and neck cancer (HNSCC), non-small cell lung cancer (NSCLC), and breast cancer (BCa) patient samples. H&E images show the morphology (A). Different cell populations were compared among different cancer types including T cells (CD3+ or CD8+), Treg (FoxP3+), and PD-L1+ tumor cells. The median of each patient from all available ROI was used for analysis (B, C). Graphs depict the mean+/-range from all patients within each cohort. **P<0.01 (1-way ANOVA).

Since Tris-EDTA buffer pH9.0 is more effective in retrieving antigens than citrate buffer pH6.0, the working concentrations of the primary antibodies needs to be reduced when choosing this buffer for retrieving antigens (Fig 1D–1I). Our method defined a standard procedure to find the optimal titration using conventional chromogenic staining for TSA-Opal IHC. The working concentration of primary antibody we obtained by incubation at 4°C overnight worked very well for all the biomarkers in our panels, as opposed to the vendor-suggested incubation conditions (e.g., CD8 incubation at RT for 30 min).

**Table 2. Patient demographics.**

|  | Number of patients (n) | Age range (years; mean±SD) | Gender (M/F) | HPV (p16) status (pos/neg/ND) |
|---|---|---|---|---|
| NSCLC | 15 | 40–81 (67.5±11.2) | 5/10 | N/A |
| HNSCC | 10 | 51–80 (63±10.1) | 7/3 | 5/3/2 |
| BCa | 12 | 35–90 (69.8±18.8) | 0/12 | N/A |

NSCLC (non-small cell lung cancer); HNSCC (head and neck squamous cell carcinoma);

BCa (breast cancer); ND (not determined); N/A (not applicable)

To facilitate panel development, we selected specific antibodies and clones that are used for clinical assessment by our Department of Pathology including CD3, CD8, CD163, and CK. The specificity and titers were tested on control FFPE tissues including human tonsil (CD3, CD8, CD163, FoxP3, and CK) and placenta (PD-L1). The proper titrations of each antibody was obtained by serial dilution tests on contiguous sections as previously described [26]. Standard controls including mouse or rabbit IgG and staing in the absence of primary or secondary antibody were used to assess antibody specificity and background. All control staining was reviewed by the pathologist (S2 Fig). The singleplex PD-L1 chromogenic stain was compared with the staining obtained from the clinical PD-L1 assay (clone sp263) and staining results were comparable [12, 27].

Previous reports noted the difference in signal amplification between conventional chromogenic and TSA-Opal IHC [12, 22, 28], however no staining protocol was provided detailing how to ensure optimal TSA-Opal staining intensity comparable with chromogenic staining. In another study, automated singleplex chromogenic staining, manual singleplex IHC, and manual mIHC staining were performed in developing 7-color mIHC panels [12, 29]. However, this protocol and the staining conditions used therein are difficult to compare as the staining conditions provided by the autostainer are different from the manual procedure. Furthermore, how one would transfer the antibody titrations used in chromogenic staining to IF stain was not addressed [12, 29]. When we evaluated the singleplex fluorescence staining on the adjacent slide to the chromogenic stain, the working concentration of primary antibodies was exactly same for each approach (Fig 2). One difference was the antibody incubation time was much shorter than the chromogenic stain, likely due to the robust signal enhancement provided by the TSA-mediated amplification, as previously reported [12, 22, 28].

For fluorescence staining, if the primary antibody was incubated overnight at 4°C, we observed that the fluorescent signal was too strong. We have optimized several additional markers using this approach and successfully transferred these to various mIHC panels. It should be noted that if one follows the vendor-suggested incubation conditions, then the titration is typically too high for TSA-Opal and can be difficult to strip, which may result in significant fluorophore overlap in subsequent mIHC panels (S3 Fig). When Tris-EDTA pH9.0 is selected for antigen retrieval, we recommended using citrate buffer pH6.0 for stripping, but not Tris-EDTA pH9.0. Heating time is the one of the most important factors which influence HIAR [12, 30]. In terms of our results of Fig 1D–1I, apparently Tris-EDTA pH9.0 is more harsh comparing with cirtrate buffer pH6.0, which is consistent with previous reports [12, 30]. This same group found that maximal retrieval using a microwave oven was obtained with a 20 min exposure for most commonly used antibodies [12, 30]. In addition multiple groups collaborated to optimizing HIAR for estrogen receptor detection and found that the optimal time for antigen retrieval was 15 to 25 min. Artifacts and diminished staining appear at 30 min after keeping a plateau [12, 31].

If choosing Tris-EDTA pH9.0 for retrieving antigens and stripping antibodies before staining the next biomarker in mIHC, excessive antigen retrieval may cause artifacts or diminished staining of some, if not all, biomarkers, after the first epitope, thus highlighting the importance of the staining sequence [12, 22]. Similarly, we do not recommend using citrate buffer pH6.0 for antigen retrieval prior to staining and stripping antibodies during the mIHC staining protocol. If we did not balance the antigen retrieval for all the biomarkers in the panel, but just followed the vendor's recommendation on the antigen retrieval such as citrate buffer pH6.0 for CD3, CD8, FoxP3 and Tris-EDTA pH9.0 for CD163, CK, PD-L1, then the individual biomarkers may exhibit different intensities when the staining sequence is changed. For example, in one study exploring 7-color mIHC protocol development, Tris-EDTA was chosen for retrieving antigens for CD3, CD4, granzyme B, and CD57 singleplex IF staining. Citrate buffer was

chosen for PD-L1 and CK staining [12, 29]. Then those antigen retrieval conditions (for all 6 biomarkers) were transferred to the mIHC protocol without balancing antigen retrieval and antibody stripping methods. Notably, when the staining sequence is changed, the intensity of staining for individual biomarkers will be altered, which may explain why the authors provided a "best sequence" of antibodies for this mIHC panel [12, 29]. However, the authors did not provide a method to resolve this problem.

Importantly, when developing a new panel with multiple biomarkers, although vendor-suggested antigen retrieval methods can be used for antibody validation, it does not mean that other antigen retrieval methods aren't suitable. Rather, these alternative methods were likely not evaluated during the original production and/or validation of the antibody. In reality, most of the antibodies effective for staining human FFPE tissue, especially those for in vitro diagnostic use, work well with Tris-EDTA pH9.0. However, there are always exceptions and if one antibody in the panel works only with citrate buffer pH6.0, then this antibody should be stained first. The same strategy works for enzyme induced epitope retrieval (EIER), though we do not recommend using this method for mIHC (manuscript in preparation).

Quantification analysis of PD-L1$^+$, FoxP3$^+$, CD3$^+$CD8$^+$, and CD3$^+$FoxP3$^+$ cell populations in human FFPE tonsil tissue verified that the staining sequence does not affect the staining intensity as long as the proper antigen retrieval method was used before mIHC. When PD-L1 was stained first by following Protocol 1, the percent of PD-L1$^+$ cells was the same or slightly lower as compared to the singleplex PD-L1 stain on the adjacent slide (Fig 4B). When FoxP3 was stained first (Protocol 2), the percent of FoxP3$^+$ cells was the same or slightly lower as compared to the singleplex FoxP3 stain on the adjacent slide (Fig 4F). The positive cell populations of these two biomarkers showed the same pattern in mIHC compared to adjacent singleplex staining. Linear Regression Analysis revealed a strong correlation between mIHC and singleplex staining (Fig 4B and 4F). When PD-L1 was stained just before DAPI by following Protocol 2, the percent PD-L1$^+$ was comparable with singleplex PD-L1 stain on the adjacent slide (Fig 4D). We obtained similar results when FoxP3 was stained just before DAPI (Protocol 1; Fig 4E). There are three potential sources of variation in these data: 1) the stripping steps during the mIHC stain may affect the final cell counts; 2) due to variable expression of PD-L1 and the potential for signals from other fluorophores to cross into the same filter, detection of cells with low staining intensity cells during image acquisition can be variable; and 3) the cell segmentation process during quantification analysis may slightly differ each time it is applied to the image, which may contribute to the variation.

When we developed these mIHC protocols, we started by testing the quality of the primary antibodies through validating the mIHC protocols with manual staining, but not with an automated system. One key finding from our study is that we have provided an easy to follow and repeatable procedure on how to get consistent and reliable staining for each individual biomarker in the mIHC panel, regardless of the staining sequence. In some published studies, automated and manual staining protocols are combined when developing the multiplex IHC protocol. Indeed, using automated equipment for standard chromogenic IHC is the standard process for most diagnostic labs. However, it is often difficult for research labs to use both approaches as automated staining platforms may not be readily accessible.

To our knowledge, few, if any, studies have examined the impact of antigen retrieval methods and staining sequence on the generation of optimal mIHC data using the TSA-based kit. Our study has focused on balancing antigen retrieval within the mIHC panel to determine which antigen retrieval buffer works best for the whole panel. We recommend using high pH solutions such as Tris-EDTA pH9.0 to retrive epitopes and citrate buffer pH6.0 to strip antibodies. This criteria can be used to develop any mIHC panel on human FFPE tissue sections as long as HIAR is required for all the biomarkers within the panel. To date, we have developed

and validated 9 mIHC panels which we are utilizing for biospecimen analysis from multiple clinical trials. Similarly, these protocols can be transitioned to an automated staining platform using these criteria. Indeed, we have successfully transferred our protocols to an autostainer. Additional antigen retrieval solutions with high pH will be tested in our lab in the near future.

In our protocols, Tris-EDTA buffer pH9.0 was used for retrieving antigens, while citrate buffer pH6.0 was used for stripping antibodies. The primary antibodies (mouse and rabbit) were used for mIHC and stained alternately. Mouse or rabbit HRP only, but not mouse and rabbit combined HRP, was used as a secondary antibody which enables 3% $H_2O_2$ to be used at RT for inactivating exogenous HRP. This helps maintain tissue integrity better than stripping in a microwave oven (2x instead of 5x stripping in our protocols) especially for high adipose tissues, such as breast tissue. This also decreases heat accumulation in the tissue section, which may cause degradation of antigens in the nuclei. It also suggests that selecting different species of the primary antibodies may be especially beneficial for some tissue types or some biomarkers when developing mIHC protocols.

To ensure exogenous HRP is completely inactivated by 3% $H_2O_2$ and that there is no carry-over signal before staining the next biomarker, we optimized the procedure by using a different Opal fluorophore for re-detection until no more signals were found [12, 15]. For example, after anti-CD3-Opal540 was stained on two serial sections of tonsil tissue following Protocol 1, 3% $H_2O_2$ incubation was performed for 10 and 30 min, respectively, at RT followed by incubation with Opal650. We found no signals were detected by Opal650 on the slide incubated with 3% $H_2O_2$ for 30 min at RT (S4 Fig). This data suggested that 30 min incubation at RT was sufficient to inactivate exogenous HRP caused by CD3 stain. A similar method was used to optimize antibody stripping steps with citrate buffer.

Also, we found that when scanning the whole slide and obtaining MSI images/ROIs, an optimized scanning protocol for each individual mIHC slide was necessary. Sharing protocols often led to overexposed or diminished signals since fluorescence (TSA-Opal) is much more sensitive than brightfield imaging. Fluorescent mIHC images allow *in situ* identification of different immune cell populations on one single section [9]. Recently, we performed 7-color mIHC on specimens from an early-stage breast cancer immunotherapy clinical trial. In this study, we verified how mIHC can be used to precisely estimate dynamic changes in tumor-infiltrating lymphocytes (TILs) score, PD-L1 expression, and other immune variables from a single FFPE section. These data helped provide insight into the spatial characteristics of the tumor microenvironment following immunotherapy treatment [27]. Our quantification analysis results revealed the differential expression of several immune cell populations across different cancer types. In this panel, PD-L1 (immune checkpoint), CD3 (T cells), CD8 (cytotoxic T cells), CD163 (histiocytes/macrophages), and FoxP3 (regulatory T cells) were analyzed in epithelial tumor (CK[+]) and surrounding stromal (CK[-]) compartments, respectively, across 3 different tumor types (Fig 5). Our initial analysis revealed significant increases in CD3[+] T cell accumulation in the stroma in NSCLC as compared to BCa specimens (Fig 5B) and increased PD-L1 expression in HNSCC tumors compared to BCa (Fig 5C). We found that a small population of CD3[-]CD8[+] cells across different tumor types. These cells could be a subpopulation of natural killer cells as CD3[-]CD8[+] and CD3[-]CD8[+]CD16[+]CD56[+] cells have been identified in peripheral blood [12, 32, 33]. To confirm this cell population, a new mIHC panel including CD3, CD8, CD16 and CD56 is under development in our lab. Future analyses are planned, including characterization of the spatial relationships between effector (CD8[+]) and suppressive (FoxP3[+] or PD-L1[+]) cell subsets across the various tumor types and correlations with clinical outcomes, such as tumor recurrence.

Currently, 8 biomarkers can be stained together using the TSA-Opal system and the advantages and disadvantages of this method were recently reviewed [34]. There is concern

regarding adding >8 markers to the panel as overheating the tissue may to excessive epitope retrieval and eventually loss of signal with consecutive rounds of staining. Thus, if additional biomarkers need to be stained on a single slide, alternative mIHC techniques are likely more appropriate, such as Imaging Mass Cytometry or use of the GeoMx Digital Spatial Profiler, which take advantage of mass spectrometry-based methods to interrogate the expression of multiple biomarkers simultaneously. These methods have disadvantages as well, including specialized equipment, high cost to stain each slide, and requirement for specialized bioinformatics support for data management and analysis.

In summary, we have investigated several critical parameters required to obtain high-quality mIHC images, particularly antigen retrieval methods. We believe these data would provide a framework that will enable more robust and consistent mIHC staining in the research lab setting.

## Supporting information

**S1 Fig. Merged and single biomarker stained by Protocol 1 and Protocol 2.**
(TIF)

**S2 Fig. CD3 (rabbit) and CK (mouse) staining with corresponding controls.**
(TIF)

**S3 Fig. High titer anti-PD-L1 stain caused signal crossover with CD3.** The circled cells are PD-L1$^+$ CD3$^-$ cells.
(TIF)

**S4 Fig. Optimization of 3% $H_2O_2$ incubation to inactivate exogenous HRP.** Incubation with 3% $H_2O_2$ for 30 min at RT was sufficient to inactivate exogenous HRP caused by CD3 stain (compare S3F vs. S3C).
(TIF)

## Acknowledgments

The authors would like to thank the Department of Pathology, Providence Portland Medical Center for providing FFPE tissue blocks, unstained slides, H&E slides, and digital slide scanning services. We also thank Linda Larocco for providing control tissue blocks and Dr. Bernie Fox for helpful discussions regarding multiplex IHC.

## Author Contributions

**Conceptualization:** Zhaoyu Sun, William L. Redmond.

**Data curation:** Zhaoyu Sun, Richard Nyberg, William L. Redmond.

**Formal analysis:** Zhaoyu Sun, Yaping Wu, Brady Bernard, William L. Redmond.

**Funding acquisition:** William L. Redmond.

**Investigation:** Zhaoyu Sun, William L. Redmond.

**Methodology:** Zhaoyu Sun, Brady Bernard, William L. Redmond.

**Project administration:** Zhaoyu Sun, William L. Redmond.

**Resources:** Zhaoyu Sun, William L. Redmond.

**Software:** Brady Bernard.

**Supervision:** William L. Redmond.

**Writing – original draft:** Zhaoyu Sun, William L. Redmond.

**Writing – review & editing:** Zhaoyu Sun, Yaping Wu, Brady Bernard, William L. Redmond.

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
