## [Decision Letter · Decision Letter 0]

1 Dec 2020

PONE-D-20-32610

Developing an Enhanced 7-color Multiplex IHC Protocol to Dissect Immune Infiltration in Human Cancers

PLOS ONE

Dear Dr. Redmond,

Thank you for submitting your manuscript to PLOS ONE. After careful consideration, we feel that it has merit but does not fully meet PLOS ONE’s publication criteria as it currently stands. Therefore, we invite you to submit a revised version of the manuscript that addresses the points raised during the review process. 

We look forward to receiving your revised manuscript.

Kind regards,

Nicole Schmitt, MD

Academic Editor

PLOS ONE

Journal Requirements:

2.Thank you for including your ethics statement:  "Primary human tumor specimens were collected following informed consent from standard-of-care surgical resection at the Providence Cancer Institute under an IRB-approved protocol (#06-108). ".   

(a) Please amend your current ethics statement to include the full name of the ethics committee/institutional review board(s) that approved your specific study.

(b) Once you have amended this/these statement(s) in the Methods section of the manuscript, please add the same text to the “Ethics Statement” field of the submission form (via “Edit Submission”).

3. In ethics statement in the manuscript and in the online submission form, please provide additional information about the patient records/samples used in your retrospective study. Specifically, please ensure that you have discussed whether all data/samples were fully anonymized before you accessed them and/or whether the IRB or ethics committee waived the requirement for informed consent. If patients provided informed written consent to have data/samples from their medical records used in research, please include this information.

4. Please note that PLOS does not permit references to “data not shown.” Authors should provide the relevant data within the manuscript, the Supporting Information files, or in a public repository. If the data are not a core part of the research study being presented, we ask that authors remove any references to these data.

5. Please include your tables as part of your main manuscript and remove the individual files. Please note that supplementary tables (should remain/ be uploaded) as separate "supporting information" files.

6.Thank you for stating the following in the Financial Disclosure section:

[This work was supported by a research grant from GlaxoSmithKline and the Providence Portland Medical Foundation (to WR). The funders had no role in the study design, data collection and analysis, decision to publish, or preparation of the manuscript.]

We note that you received funding from a commercial source: [GlaxoSmithKline]

We note that you have a patent relating to material pertinent to this article. Please provide an amended statement of Competing Interests to declare this patent (with details including name and number), along with any other relevant declarations relating to employment, consultancy, patents, products in development or modified products etc. Please confirm that this does not alter your adherence to all PLOS ONE policies on sharing data and materials, as detailed online in our guide for authors http://journals.plos.org/plosone/s/competing-interests by including the following statement: "This does not alter our adherence to  PLOS ONE policies on sharing data and materials.” If there are restrictions on sharing of data and/or materials, please state these. Please note that we cannot proceed with consideration of your article until this information has been declared.

Reviewers' comments:

Reviewer's Responses to Questions

**Comments to the Author**

1. Is the manuscript technically sound, and do the data support the conclusions?

Reviewer #1: Yes

Reviewer #2: Partly

2. Has the statistical analysis been performed appropriately and rigorously? 

Reviewer #1: Yes

Reviewer #2: No

3. Have the authors made all data underlying the findings in their manuscript fully available?

Reviewer #1: Yes

Reviewer #2: Yes

4. Is the manuscript presented in an intelligible fashion and written in standard English?

Reviewer #1: Yes

Reviewer #2: Yes

5. Review Comments to the Author

Reviewer #1: This is a well conceived project that aims to optimize multiplex IHC staining of tumor specimens. The protocols are technically sound and very valuable additions to the existing literature. The workflow is described in details and clear to follow. There are a couple of minor comments that may merit further consideration as this pipeline is advanced.

1. A challenge with multiplex IHC quantitation is its consistency across pathologists/institutions. Would the authors be able to recommend a consistent score for each marker for positivity/threshold calling? Could this be built into an SOP for each marker together with the other components of this workflow?

2. A discussion on the workflow details to expand existing panels would be helpful.

Overall, this is an excellent study that has produced carefully designed protocols to address many challenges in the current multiplex IHC practice. It is a highly valuable addition to the existing literature.

Reviewer #2: General notes:

The manuscript is a well-written paper easy to understand. The authors aimed first to develop a multiplexing assay for markers used to analyze infiltrating lymphocytes in tumors. However, I am struggling to understand the real scientific value of the work presented. In the introduction, this might be presentation of the ideal set of antibodies used in the panel. Despite the potential limitation as to the novelty of this work, The authors do provide a succinct method for multiplex IHC using a commercially available system and evidence of robust optimization parameters. Ultimately, they show that sequence of multiplexing does not affect the quality or quantification of the IF. The present several iterations of comparisons and then test their method in new carcinoma tissue (HNSCC, BCC, and NSCLC) samples to verify their method. Overall, the quality of the figures is good, but some details are lacking that are required for more robust analysis of their data. I think this provides an easy to follow roadmap for mIHC which could be applied by both cancer researchers and immunologists alike, given proper multispectral imaging capabilities.

Typographical and Technical issues:

P10 L191: “First”

P17 L348 reference: Shi et al. Need proper format.

P18 L366 reference: Gorris et al. Need proper format

P19 L383: Why chose 15 ROIs and not quantify the entirety of the slide if they are serial sections? If you chose ROIs, then please show the correlation matrix for each of the stains across each of the ROIs. How were the ROIs selected? Please present the statistics for this relationship - and for all other panels presented.

P22 L461: Please show this data as supplemental data. I think because of the prominence as a methods manuscript it is only useful to show some of these optimization data as a supplement.

Figure 1. Quantification of the % positive by marker type by both methods. You could put the XX%+/- SD on each slide in the bottom right corner, or show dot-plot/box/whisker plots as you do in Figure 5, 6 (best). I would suggest quantifying the entire slide using qupath/visiopharm/halo/vectra and then present representative photomics – surely the authors have access to these capabilities.

Figure 2. Same critique as Figure 1. Chromogenic vs Opal: Validation of single-plex Opal fluorescent stains in human FFPE tonsil tissue specimens. Show this statistically across the markers validated. This is important because it shows overall the data loss/gain by methodology.

Figure 3. Same critique as Figure 1. However, I think that you could limit the figures to only the multiplex IHC figures for protocol 1 and 2 and show that the protocol makes no difference in percent positive by cell type or marker. I would move the single-plex to supplemental and the quantification put in the main.

Figure 4. ONLY include the most relevant mIF here and move the rest of the IF to supplemental. For the line graphs, this is not the ideal way to present this data (an x-y scatter for each marker?). There should be pretty decent correlation if each line represents the expression percent for each ROI, however what is NOT clear is what marker they are referring to. I think you could show the correlation matrix to show the correlations for each marker b/w single- and multiple-plex. I would also suggest investigating and reporting on the colocalization of CD3-CD8 and CD3-FOXP3 relationships (and other cell-type classification relationships) by sequence are not affected dramatically - that is, define more clearly how mIHC can show granular differences in cellularity/inflammation in given tissue sections.

The statement on page 3 L59 to p4 L61-62 needs a reference. Several works address this (just a few of these are: DOI: 10.1369/jhc.2009.953240, DOI: 10.4049/jimmunol.1800878 . Also, previous work reports a similar panel with much details in methods and all the necessary controls DOI: 10.1038/s41598-017-13942-8 other publications on the validation of the technique in:

DOI: 10.1016/j.ymeth.2014.08.016; DOI: 10.1369/jhc.6A7134.2007 and

DOI: 10.1016/0022-1759(92)90073-3 to mention just a few. The authors should discuss how their work differs from other publications and analyze their work's advantages and disadvantages.

The idea of replacing half of the stripping with peroxidase inhibitor step instead of stripping buffer seems plausible and interesting to explore. However, this approach is not entirely validated, the authors do not study the carryover signal and do not test for NON-specific signal properly. As control they report is buffer, when the more appropriate control is used normal IgG (this needs to be shown) in the primary animal at the same concentration used in the primary.

The development of panels aimed to do similar analysis had been performed and reported previously.

In summary i think that the technical contribution is minor if not minimal. Also to properly validate what they proposing, the authors are are missing two major control (signal carry over and unspecific signal).

6. PLOS authors have the option to publish the peer review history of their article (what does this mean?). If published, this will include your full peer review and any attached files.

Reviewer #1: **Yes: **Yu Leo Lei

Reviewer #2: No

---

## [Author Response · Author response to Decision Letter 0]

1 Feb 2021

Response to Reviewers' Comments

Sun Z et al., PLOS ONE Manucript #PONE-D-20-32610

Reviewer #1

1. A challenge with multiplex IHC quantitation is its consistency across pathologists/institutions. Would the authors be able to recommend a consistent score for each marker for positivity/threshold calling? Could this be built into an SOP for each marker together with the other components of this workflow?

• We appreciate the reviewer’s comment and agree that this is a very important issue. Conventional singleplex chromogenic staining has been recognized as the gold standard for IHC. Pathologists have relied on this staining technique to score patient samples in clinic (e.g., PD-L1). However, the staining intensity of each individual biomarker is determined by multiple factors including antibody species and clone, tissue fixation, antigen retrieval methods, fresh vs. archive tissue, tissue type, type of IHC staining equipment, etc. For example, different automated IHC staining platforms such as Leica Bond Automated IHC Stainer vs. Ventana BenchMark, which are widely used clinically, can have differences in staining intensity. Thus, it can be challenging to obtain consistent scores for mIHC for each individual biomarker across pathologists and/or institutions. But as long as appropriate “universal controls” are set up between these two staining systems, it is feasible to get consistent scores. There are some limitations, however, as based upon our knowledge the Ventana internal control, for example, is only available for the Ventana BenchMark system. 

• For most research labs, manual staining is used as the main approach for IHC. However, antigen retrieval methods, including buffers and equipment (e.g., microwave, pressure cooker, autoclave, and water bath) are variable from lab to lab. These issues can make it difficult to obtain consistent scores for the same biomarker across institutions. To develop an optimized SOP, a proper antigen retrieval method has to be decided before developing the protocol. Fresh FFPE tissue sections are required. To obtain the appropriate score, manual stained fresh FFPE tissue sections need to be compared with automated stained sections (since some of the scoring has been approved by FDA or validated by clinic) by pathologists. This is critical especially for clinical trial projects. Then the former can be used as a control to score the same biomarker in the same type of tissue sections stained manually by following the optimized SOP. 

• Setting up positivity/thresholds for each biomarker is complex. For example, the range of PD-L1 expression is rather large. The clone of anti-PD-L1 used in the most research lab is E1L3N while clinical labs use different clones including SP263, SP142, 22C3, and 28-8. These differences made defining the threshold of PD-L1 challenging. We compared each of the clones used in clinic with E1L3N and found that it was comparable with SP263. Therefore, the pathologist can decide the threshold by the parallel chromogenic stain of these two clones. When possible, we select the same clones used for clinical IHC for use in our biomarker mIHC panels (such as Her2). 

2. A discussion on the workflow details to expand existing panels would be helpful.

• We have developed 9 human mIHC protocols using the same strategy since 2017. We have tested and validated these protocols with specimens from various clinical trials at our Institute (Sanchez K. et al., Breast Cancer Research, 2021, doi: 10.1186/s13058-020-01378-4). See lines 560-561.

Reviewer #2 

1. The authors aimed first to develop a multiplexing assay for markers used to analyze infiltrating lymphocytes in tumors. However, I am struggling to understand the real scientific value of the work presented. In the introduction, this might be presentation of the ideal set of antibodies used in the panel. Despite the potential limitation as to the novelty of this work, the authors do provide a succinct method for multiplex IHC using a commercially available system and evidence of robust optimization parameters. Ultimately, they show that sequence of multiplexing does not affect the quality or quantification of the IF. The present several iterations of comparisons and then test their method in new carcinoma tissue (HNSCC, BCC, and NSCLC) samples to verify their method. Overall, the quality of the figures is good, but some details are lacking that are required for more robust analysis of their data. 

• We appreciate the Reviewer’s remarks. We recently published additional validation of a 7-color mIHC panel that was performed on human FFPE breast cancer tissue using six biomarkers (CD3, CD8, CD163, PD-L1, Foxp3, and cytokeratin) plus DAPI using Protocol 1 (Sanchez K. et al., Breast Cancer Research, 2021, doi: 10.1186/s13058-020-01378-4). In this study, we verified that mIHC can be used to precisely estimate dynamic changes in TIL score, PD-L1 expression, and other immune variables from a single FFPE section, and therefore to help understand spatial and comprehensive characterization of novel cancer immunotherapy drugs (see lines 588-593).

2. Typographical and technical issues:

P10 L191: “First”

P17 L348 reference: Shi et al. Need proper format.

P18 L366 reference: Gorris et al. Need proper format.

• We apologize for these errors and have made all the indicated corrections. 

3. P19 L383: Why chose 15 ROIs and not quantify the entirety of the slide if they are serial sections? If you chose ROIs, then please show the correlation matrix for each of the stains across each of the ROIs. How were the ROIs selected? Please present the statistics for this relationship - and for all other panels presented.

• The 7 color (6 biomarkers plus DAPI) mIHC slides were stained using TSA-Opal kit and scanned with a Vectra 3 Automated Quantitative Pathology Imaging system. The standard workflow of imaging is to get whole slide scanned with low power first and then choose Multiple Spectral Images (MSIs) or Regions of Interest (ROIs) on the whole slide scanned image. The Vectra system has 5 filters. The wavelength of fluorescent colors (Opal dyes) is mixed. That means more than one fluorescent color can go through the same one filter when imaging. To differentiate each individual fluorophore/biomarker and get rid of autofluorescence caused by tissue fixation, unmixed spectra need to be extracted using a spectral library. This step is allowed to be finished on MSI images, but not on whole slide scanned images using InForm Advanced Image Analysis Software. This limits the whole slide 7-color mIHC analysis using either Qupath Quantitative Pathology & Bioimage Analysis or InForm Advanced Image Analysis Software. 

• The workflow we set up is using two adjacent tissue sections: one H&E and one 7-color mIHC stain. The pathologist in our team reviews the H&E slide and circles the target area such as tumor or tumor-stoma interface. Then we choose MSI images cross the circled area on the adjacent mIHC section. That’s why we include H&E images in Figures 3, 5, and Supplemental Figure 1. We normally select 10-15 MSI images depending upon the size of tissue biopsy and select ROIs across the whole section, whenever possible. 

4. P22 L461: Please show this data as supplemental data. I think because of the prominence as a methods manuscript it is only useful to show some of these optimization data as a supplement.

• We have added this as requested. Please see Supplemental Figure 3.

5. Figure 1. Quantification of the % positive by marker type by both methods. You could put the XX%+/- SD on each slide in the bottom right corner or show dot-plot/box/whisker plots as you do in Figure 5, 6 (best). I would suggest quantifying the entire slide using qupath/visiopharm/halo/vectra and then present representative photomics – surely the authors have access to these capabilities.

• We completed the quantification of % for each biomarker using Qupath software and have added those data (mean percent � SD) in Figure 1D-I (see lines 172-174 and 347-350).

6. Figure 2. Same critique as Figure 1. Chromogenic vs Opal: Validation of singleplex Opal fluorescent stains in human FFPE tonsil tissue specimens. Show this statistically across the markers validated. This is important because it shows overall the data loss/gain by methodology.

• We have added the requested data (mean percent � SD) in Figure 2 (see lines 367-370). 

7. Figure 3. Same critique as Figure 1. However, I think that you could limit the figures to only the multiplex IHC figures for protocol 1 and 2 and show that the protocol makes no difference in percent positive by cell type or marker. I would move the singleplex to supplemental and the quantification put in the main.

• We have move the singleplex image to Supplemental Figure 1 and relevant analysis is shown in Figure 4. 

8. Figure 4. ONLY include the most relevant mIF here and move the rest of the IF to supplemental. For the line graphs, this is not the ideal way to present this data (an x-y scatter for each marker?). There should be pretty decent correlation if each line represents the expression percent for each ROI, however what is NOT clear is what marker they are referring to. I think you could show the correlation matrix to show the correlations for each marker b/w single- and multiplex. I would also suggest investigating and reporting on the colocalization of CD3-CD8 and CD3-FOXP3 relationships (and other cell-type classification relationships) by sequence are not affected dramatically - that is, define more clearly how mIHC can show granular differences in cellularity/inflammation in given tissue sections.

• We appreciate this comment and have conducted regression analysis including CD3+CD8+ and CD3+Foxp3+ cells populations (Figs. 4B, 4D, 4E, 4F, 4G, and 4H), which shows significant correlations (see lines 413-417).

9. The statement on page 3 L59 to p4 L61-62 needs a reference. Several works address this (just a few of these are: DOI: 10.1369/jhc.2009.953240, DOI: 10.4049/jimmunol.1800878. 

• We have added these references to page 3 (see line 60).

10. Also, previous work reports a similar panel with much details in methods and all the necessary controls DOI: 10.1038/s41598-017-13942-8 other publications on the validation of the technique in: DOI: 10.1016/j.ymeth.2014.08.016; DOI: 10.1369/jhc.6A7134.2007 and DOI: 10.1016/0022-1759(92)90073-3 to mention just a few. The authors should discuss how their work differs from other publications and analyze their work's advantages and disadvantages.

• We appreciate the Reviewer’s comments. Regarding Parra et al. (DOI: 10.1038/s41598-017-13942-8), in this study the authors validated sequential mIHC staining using the TSA-Opal kit. Automated and manual staining were combined in this study. The protocol of each individual antibody validation, such as the selection of antigen retrieval method on the autostainer was not addressed. To our knowledge, both citrate buffer and Tris-EDTA buffer can be used on Leica Bond-Max. However, we believe that this step can require additional testing and validation and may be more difficult to follow for researchers without experience running autostainers or for those new to the complexities of mIHC. 

• In Parra et al., after antibody validation was performed on Leica Bond-Max, singleplex and mIHC were performed manually. As we discussed in our manuscript, the staining conditions provided by automated equipment are different from manual staining especially with respect to antigen retrieval. In Parra et al., the authors did not address the correlation of protocol development between singleplex chromogenic stain and IF stain nor how to decide the antibody dilution factor or antigen retrieval method in singleplex IF. There were also differences in antigen retrieval buffers selected by the authors (Tris-EDTA vs. citrate buffer) that were altered from what is indicated by the Ab data sheet from the vendor, which was not explained by the authors. We believe that the authors did fully appreciate the complexities and/or impact of transferring a specific antigen retrieval/stripping method from singleplex IHC to sequential mIHC. This is one reason why the staining intensity of individual biomarkers can change when the staining sequence is altered. Overall, their study realized the staining sequence affects the staining intensity of individual biomarkers using the protocol they developed, but did not provide a proper approach to resolve this problem nor a strategy on how to properly develop a mIHC panel using the TSA-Opal kit (see lines 476-481 and 509-517).

• Regarding DOI: 10.1016/j.ymeth.2014.08.016, we have cited this work and have used the same validation method in our study. Singleplex IHC has been used as a control to assess the same biomarkers in the adjacent mIHC slide (see lines 50, 60, 589).

• With respect to DOI: 10.1369/jhc.6A7134.2007, we used a similar method as published to verify whether the stripping is clean enough before we stain the next biomarker. Before we stain Foxp3 following Protocol 1, we need to inactivate exogenous HRP which was caused by CD3-Opal540 staining. We did 10- and 30-min incubations with 3% H2O2 at RT on two serial FFPE tonsil sections, respectively. Then these two sections were re-detected with Opal650. We could see CD3-Opal650 signals on the section with 10 min 3% H2O2 incubation, but not on the section with 30 min incubation (Supplemental Figure 4). This data demonstrates that the latter is long enough to remove exogenous HRP. A similar method was used in stripping steps with citrate buffer. The comparison with singleplex IF stain verified that each stripping step is properly performed (see lines 575-583 and Supplemental Figure 4).

• Finally, regarding DOI: 10.1016/0022-1759(92)90073-3, to ensure the proper staining of each individual biomarker, we have chosen the same antibody clones (CD3, CD8, CD163, and cytokeratin) as used by our Clinical Pathology Laboratory in the Department of Pathology at Providence Health and Services, Portland OR. The specificity and titrations were tested on control tissue (human FFPE tonsil) for CD3, CD8, CD163, Foxp3 and cytokeratin and placenta for validation of PD-L1 staining. Standard controls including mouse or rabbit IgG and staining in the absence of primary or secondary antibody were arranged and slides were reviewed by a board-certified pathologist (Supplemental Figure 2). The singleplex PD-L1 (clone E1L3N) chromogenic stain was compared with the clone (sp263) which has been used in clinical testing. The staining pattern has very high consistency (Sanchez K. et al., Breast Cancer Research, 2021, doi: 10.1186/s13058-020-01378-4) (see lines 464-472).

11. The idea of replacing half of the stripping with peroxidase inhibitor step instead of stripping buffer seems plausible and interesting to explore. However, this approach is not entirely validated, the authors do not study the carryover signal and do not test for NON-specific signal properly. As control they report is buffer, when the more appropriate control is used normal IgG (this needs to be shown) in the primary animal at the same concentration used in the primary.

• We have added as a Supplemental Figure 4 to show the study of carryover signals (see lines 575-583). The specificity of each individual biomarker and signal carryover (or lack thereof) was verified by comparing with singleplex IHC on the adjacent slide as in the previous published study (doi: 10.1016/j.ymeth.2014.08.016). Normal mouse or rabbit IgG at the same concentration used in the primary has been used as a control (see Supplemental Figure 2).

12. The development of panels aimed to do similar analysis had been performed and reported previously.

• We respectfully disagree with the reviewer as, to our knowledge, only one published study used Tris-EDTA pH9.0 for antigen retrieval and citrate buffer for antibody stripping in developing a mouse mIHC panel (DOI: 10.4049/jimmunol.1800878). As we detailed above, there are significant differences in our approach and we believe that this manuscript represents an important resource for our colleagues developing mIHC panels that will help foster improved staining and allow for the acquisition of higher quality imaging data. 

13. … to properly validate what they proposing, the authors are missing two major controls (signal carry over and unspecific signal).

• Mouse or rabbit IgG has been used for testing antibody specificity (See Supplemental Figure 2). Since we work closely with our Pathology Department, we selected clinically validated clones where feasible. Singleplex IHC was verified by comparing with the gold-standard chromogenic stain from our clinical lab and verified by the pathologist. Signal carryover was verified by re-detecting with a different Opal dye (see Supplemental Figure 4) and comparing with singleplex IHC (Figure 4A-F) on the adjacent slide as detailed above.

---

## [Editor Report · Decision Letter 1]

4 Feb 2021

Developing an Enhanced 7-color Multiplex IHC Protocol to Dissect Immune Infiltration in Human Cancers

PONE-D-20-32610R1

Dear Dr. Redmond,

We’re pleased to inform you that your manuscript has been judged scientifically suitable for publication and will be formally accepted for publication once it meets all outstanding technical requirements.

Kind regards,

Nicole Schmitt, MD

Academic Editor

PLOS ONE

Additional Editor Comments (optional): Thank you for carefully addressing all of the reviewers' concerns. I agree that this manuscript may be of help to others hoping to develop their own mIHC panels.

---

## [Editor Report · Acceptance letter]

5 Feb 2021

PONE-D-20-32610R1 

Developing an Enhanced 7-color Multiplex IHC Protocol to Dissect Immune Infiltration in Human Cancers 

Dear Dr. Redmond:

I'm pleased to inform you that your manuscript has been deemed suitable for publication in PLOS ONE. Congratulations! Your manuscript is now with our production department. 

Kind regards, 

on behalf of

Dr. Nicole Schmitt 

Academic Editor

PLOS ONE